# AutoPrivacy: Automated Layer-wise Parameter Selection for Secure Neural Network Inference

Qian Lou
Indiana University Bloomington
louqian@iu.edu

Song Bian
Kyoto University
sbian@easter.kuee.kyoto-u.ac.jp

Lei Jiang
Indiana University Bloomington
jiang60@iu.edu

## Abstract

Hybrid Privacy-Preserving Neural Network (HPPNN) implementing linear layers by Homomorphic Encryption (HE) and nonlinear layers by Garbled Circuit (GC) is one of the most promising secure solutions to emerging Machine Learning as a Service (MLaaS). Unfortunately, a HPPNN suffers from long inference latency, e.g., $\sim 100$ seconds per image, which makes MLaaS unsatisfactory. Because HE-based linear layers of a HPPNN cost 93% inference latency, it is critical to select a set of HE parameters to minimize computational overhead of linear layers. Prior HPPNNs over-pessimistically select huge HE parameters to maintain large noise budgets, since they use the same set of HE parameters for an entire network and ignore the error tolerance capability of a network.

In this paper, for fast and accurate secure neural network inference, we propose an automated layer-wise parameter selector, AutoPrivacy, that leverages deep reinforcement learning to automatically determine a set of HE parameters for each linear layer in a HPPNN. The learning-based HE parameter selection policy outperforms conventional rule-based HE parameter selection policy. Compared to prior HPPNNs, AutoPrivacy-optimized HPPNNs reduce inference latency by $53\% \sim 70\%$ with negligible loss of accuracy.

## 1   Introduction

Machine Learning as a Service (MLaaS) is an emerging computing paradigm that uses powerful cloud infrastructures to provide machine learning inference services to clients. However, in the setting of MLaaS, cloud servers can arbitrarily access input and output data of clients, thereby introducing privacy risks. Privacy is important when clients upload their sensitive information, e.g., healthcare records and financial data, to cloud servers. Recent works [1, 2, 3, 4, 5] create Hybrid Privacy-Preserving Neural Networks (HPPNNs) to achieve both low inference latency and high accuracy using a combination of Homomorphic Encryption (HE) and Garbled Circuit (GC). Particularly, DELPHI [5] obtains the state-of-the-art inference latency and accuracy through implementing linear layers by HE, and computing activation layers by GC. However, HPPNNs still suffer from long inference latency. For instance, inferring **one** single CIFAR-10 image by DELPHI ResNet-32 [5] costs $\sim 100$ seconds and has to exchange 2GB data. Particularly, the HE-based linear layers of DELPHI cost 93% of its inference latency, thereby becoming its performance bottleneck.

The computational overhead of HE-based linear layers in prior HPPNNs is decided by their HE *parameter*s including the plaintext modulus $p$, the ciphertext modulus $q$, and the polynomial degree $n$. HE enables homomorphic additions and multiplications on ciphertexts by manipulating polynomials whose total term number and coefficients are defined by $p$, $q$ and $n$. Each HE operation introduces

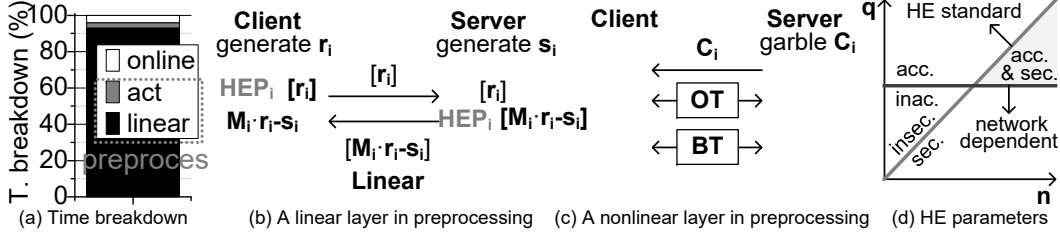

(a) Time breakdown     (b) A linear layer in preprocessing     (c) A nonlinear layer in preprocessing     (d) HE parameters

Figure 1: The bottleneck analysis, working flow and HE parameter selection of DELPHI.

a small noise. Decrypting a HE output may have errors, if the total noise accumulated along a HE computation path exceeds the noise budget decided by $p$, $q$ and $n$. Fully HE adopts *bootstrapping* operations to eliminate noises, and thus is not sensitive to noise budget. However, to avoid extremely slow bootstrapping operations of fully HE, prior HPPNNs use leveled HE that allows only a limited noise budget. A large noise budget requires large $p$, $q$ and $n$, significantly increasing computational overhead of polynomial additions and multiplications.

In order to keep a fixed (and extremely small) decryption failure rate for correct ciphertext decryption, prior HPPNNs over-pessimistically assume huge noise budgets using large $p$, $q$ and $n$. First, prior HPPNNs do not consider the error tolerance of neural networks when defining their HE parameters $p$, $q$ and $n$. We found that a HPPNN can tolerate some decryption errors without degrading private inference accuracy. Second, prior HPPNNs assume the same $p$, $q$ and $n$ for all layers. Different layers in a neural network have different architectures, e.g., weight kernel size and output channel number, and thus different error tolerances. Therefore, assuming the same worst case HE parameters for all layers substantially increases the computational overhead of a HPPNN. However, defining a set of $p$, $q$ and $n$ for each layer via hand-crafted heuristics is so complicated that even HE and machine learning experts may obtain only sub-optimal results. In this paper, we propose an automated layer-wise HE parameter selection technique, AutoPrivacy, for fast and accurate HPPNN inferences.

## 2 Background and Motivation

**Threat Model**. Our threat model is the same as that in [5]. AutoPrivacy is designed for the two-party semi-honest setting, where only one of the parties may be corrupted by an adversary. Both parties adhere the security protocol, but try to learn information about private inputs of the other party from messages they receive. AutoPrivacy aims to protect the client's privacy, but does not prevent the client from learning the architecture of the neural network used by the server [5].

**Privacy-Preserving Neural Network**. Prior HPPNNs [1, 2, 3, 4] combine Homomorphic Encryption (HE) and Garbled Circuit (GC) to support privacy-preserving inferences. An HPPNN inference includes a preprocessing stage and an online stage. During a preprocessing stage, a server and a client prepare Beaver's triples, setup HE parameters, and perform oblivious transfers (OT) for the next online stage. In the online stage, the server and the client jointly compute to obtain the inference result. As Figure 1(a) shows, the preprocessing stage for activation and linear layers dominates HPPNN inference latency in one of the best performing recent works [5]. The security protocol of the preprocessing stage for [5] is summarized as follows.

- *HE-based linear layer*. Linear layers in a HPPNN are either implemented through HE or Beaver's triples generated by HE. HE enables homomorphic computations on ciphertexts without decryption. Given a public key $pk$, a secret key $sk$, an encryption function $\epsilon()$, and a decryption function $\sigma()$, $\times$ is a homomorphic operation, if there is another operation $\otimes$ such that $\sigma(\epsilon(x_1, pk) \otimes \epsilon(x_2, pk), sk) = \sigma(\epsilon(x_1 \times x_2, pk), sk)$, where $x_1$ and $x_2$ are plaintexts. Although most HE schemes, e.g., BFV [6], can support fast matrix-vector multiplications with SIMD evaluation, HE-based linear layers are still the performance bottleneck of a HPPNN. As Figure 1(a) shows, HE-based linear layers consume 71% of inference latency of the latest HPPNN DELPHI [3]. During the preprocessing stage of a linear layer ($L_i$), the client and the server generate two masking vector $r_i$ and $s_i$ respectively for $L_i$, as shown in Figure 1(b). The client encrypts $r_i$ as $[r_i]$, and sends $[r_i]$ to the server, while the server homomorphically computes $[M_i \cdot r_i - s_i]$ and sends it to the client, where $M_i$ indicates the weights and biases of $L_i$. The client decrypts $[M_i \cdot r_i - s_i]$. The server holds $s_i$, so the client and the server hold an additive secret sharing of $M_i \cdot r_i$.

- *GC-based nonlinear layer*. Prior HPPNNs implement nonlinear layers by GC that is a cryptographic protocol enabling the server and the client to jointly compute a nonlinear layer over their private data without learning the other party's data. In GC, an activation is represented by a Boolean circuit.

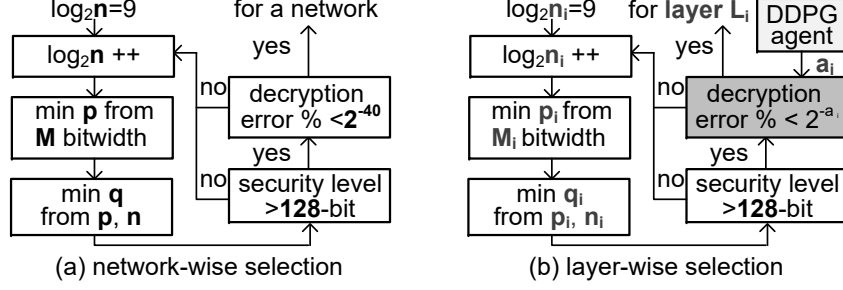

Figure 2: AutoPrivacy: (a) HE parameter generation of prior HPPNNs; and (b) HE parameter generation of AutoPrivacy.

As Figure 1(c) shows, the server firstly garbles an activation, generates its garbled table ($C_i$), and sends it to the client. The client receives $C_i$ by Oblivious Transfer (OT) [3]. In the online stage, the client evaluates $C_i$ to produce an activation result.

- *Beaver's-Triples-based activations.* The latest HPPNN DELPHI [3] also adopts Beaver's Triples (BT) to implement quadratic approximations of the activation function to further reduce computing overhead of GC-based nonlinear layers. To maintain the same inference accuracy, DELPHI uses both GC- and BT-based activations in its nonlinear layers.

Compared to GC-only-based neural networks, e.g., DeepSecure [7], and HE-only-based neural networks, e.g., CryptoNets [8], SHE [9], and Lola [10], HPPNNs [5] decrease inference latency by $\sim 100\times$ and improve inference accuracy by $1\% \sim 4\%$.

**The BFV Cryptosystem and Its HE Parameters**. By following DELPHI [5], we adopt BFV [11] to implement HE operations in a HPPNN. We use $[r]$ to indicate a ciphertext holding a message vector $r$, where the plaintext is a ring $\mathcal{R}_p = \mathbb{Z}_p[x]/(x^n+1)$ with plaintext modulus $p$ and cyclotomic order $n$. In BFV, due to its packing technique, a ciphertext $[r] \in \mathcal{R}_q^2$ is a set of two polynomials in a quotient ring $\mathcal{R}_q$ with ciphertext modulus $q$. For the encryption of a packed polynomial $m$ containing the elements in $r$, a BFV ciphertext is structured as a vector of two polynomials $(c_0, c_1) \in \mathcal{R}_q^2$. Specifically,

$$c_0 = -a \qquad (1) \qquad\qquad c_1 = a \cdot s + \frac{q}{p}m + e_0 \qquad (2)$$

where $a$ is a uniformly sampled polynomial, while $s$ and $e_0$ are polynomials whose coefficients drawn from $\mathcal{X}_\sigma$, where $\sigma$ is the standard deviation. The decryption simply computes $\frac{p}{q}(c_0 s + c_1) = m + \frac{p}{q}e_0$. When $\frac{q}{p} \gg e_0$, $e_0$ can be removed. As Figure 1(d) exhibits, the larger $q$ is, the more likely $e_0$ can be omitted, the more accurate the BFV cryptosystem is. For each group of $p$, $q$, $n$ and $\sigma$, the LWE-Estimator [12] can estimate the BFV security level $\lambda$ based the BFV standard. The larger $q$ and $n$ are, the more secure a BFV-based cryptosystem is. To guarantee the correctness and execution efficiency of BFV, the HE parameters have to follow the **5 rules** [3]: ❶ $n$ is a power of two; ❷ $q \equiv 1 \bmod n$; ❸ $p \equiv 1 \bmod n$; ❹ $|q \bmod p| \approx 1$; and ❺ $q$ is pseudo-Mersenne.

**Batching**. To support SIMD, BFV [11] adopts the Chinese Remainder Theorem (CRT) to pack $n$ integers modulo $p$ into one plaintext polynomial $m$. BFV batching only works when $p$ is a primer number and congruent to 1 (mod $2n$). This $p$ assures that there exists a primitive $2n$-th root of unity $\zeta$ in the integer domain modulo $p$ ($\mathbb{Z}_p$), so that polynomial modulus $x^n + 1$ factors modulo $p$ as:

$$x^n + 1 = (x - \zeta)(x - \zeta^3)...(x - \zeta^{2n-1})(mod\, p). \qquad (3)$$

CRT offers an isomorphism ($\cong$) between a plaintext polynomial $m \in \mathcal{R}_p$ and $n$ integers $\prod_{i=0}^{n-1} \mathbb{Z}_p$:

$$\mathcal{R}_p = \frac{\mathbb{Z}_p[x]}{x^n + 1} = \frac{\mathbb{Z}}{\prod_{i=0}^{n-1} x - \zeta^{2i+1}} \cong \prod_{i=0}^{n-1} \frac{\mathbb{Z}_p[x]}{x - \zeta^{2i+1}} \cong \prod_{i=0}^{n-1} \mathbb{Z}_p[\zeta^{2i+1}] \cong \prod_{i=0}^{n-1} \mathbb{Z}_p. \qquad (4)$$

Based on the BFV batching, $n$ coefficient-wise additions or multiplications in integers modulo $p$ are computed by one single addition or multiplication in $\mathcal{R}_p$.

**HE Parameter Selection**. Prior HPPNNs [1, 2, 5, 3, 4] decide their HE parameters using the flow shown in Figure 2(a). For an entire neural *network*, prior HPPNNs first choose the cyclotomic order $n$ that is a power of two and typically $\geq 10^{10}$, and then select a prime $p \geq M$, where $M$ is the maximum plaintext value of the neural network model (i.e., weights and biases). Prior HPPNNs must

| HPPNN | plaintext modulus $\log(p)$ | cyclotomic order $\log(n)$ | ciphertext modulus $\log(q)$ | standard deviation $\sigma$ | security level $\lambda$ | decryption error % |
|---|---|---|---|---|---|---|
| DELPHI [5] | 22 | 13 | 180 | 3.2 | > 128 | > $2^{-40}$ |
| DARL [4] | 14 | 13 | 165 | 3.2 | > 128 | $2^{-40}$ |

Table 1: The HE parameters of prior HPPNNs.

guarantee $p \equiv 1 \mod n$, otherwise they increase $p$. By the LWE-Estimator [12], based on $n$, $p$, a standard deviation $\sigma$ of noise and a security level value $\lambda$ (e.g., 128-bit), prior HPPNNs compute the maximum value ($q_{max}$) of $q$. According to the network architecture, prior HPPNNs obtain the minimal value ($q_{min}$) of $q$ that makes the HE decryption failure rate smaller than $2^{-40}$ [4]. From $q_{min}$ to $q_{max}$, prior HPPNNs choose the smallest $q$ that can meet the other constraints imposed by the 5 rules of HE parameters. A recent compiler [13] implements the procedure of HE parameter selection shown in Figure 2(a) for a neural network. To provide circuit privacy, prior HPPNNs [5] have to implement *noise flooding* [5] by tripling $\log_2(q)$ and quadrupling $n$. The HE parameters of recent HPPNNs are shown in Table 1.

**HE Execution Efficiency**. The latency of HE-based linear layers of a HPPNN is decided by its HE parameters, i.e., $n$ and $q$. Inputs of a HPPNN are encrypted into polynomials consisting of $n$ terms. Homomorphic multiplications during a HPPNN inference are performed through polynomial multiplications, where the coefficient of each term has a modulus of $q$. BFV [11] adopts the Number-Theoretic Transform (NTT) [14] with modular reduction to accelerate polynomial multiplications. The time complexity of a NTT-based polynomial multiplication is $\mathcal{O}(n \log n)$. Because $q$ can be larger than 64-bit, recent BFV implementations use the Residue Number System (RNS) [6] to decompose large $q$ into vectors of smaller integers. A smaller $q$ greatly accelerates HE operations. As Figure 3 shows, $2 \times n$ and $1.5 \times \log(q)$ increases the latency of a HE multiplication by $3.2\times$.

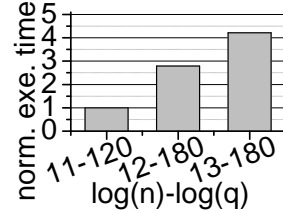

Figure 3: The latency comparison of a HE multiplication with varying $n$ and $q$ (normalized to 11-120, where $\log(n)$=11 and $\log(q)$=120).

**Drawbacks of Prior HE Parameter Selection Policies**. We find prior HPPNNs over-pessimistically choose huge values of $n$ and $q$, resulting in unnecessarily long privacy-preserving inference latency. First, prior HPPNNs ignore their error tolerance capability, i.e., a HPPNN encrypted with smaller $n$ and $q$ producing a higher decryption error rate may still achieve the same inference accuracy but use much shorter inference latency. Second, different layers of a HPPNN have distinctive architectures, and thus can tolerate different amounts of decryption errors. So a HPPNN should select $p$, $n$ and $q$ for each layer to shorten its inference latency. Choosing $p$, $n$ and $q$ for each layer does not expose more information to the client, since prior HPPNNs [1, 2, 5, 3, 4] cannot protect the architecture of the network from being known by the client.

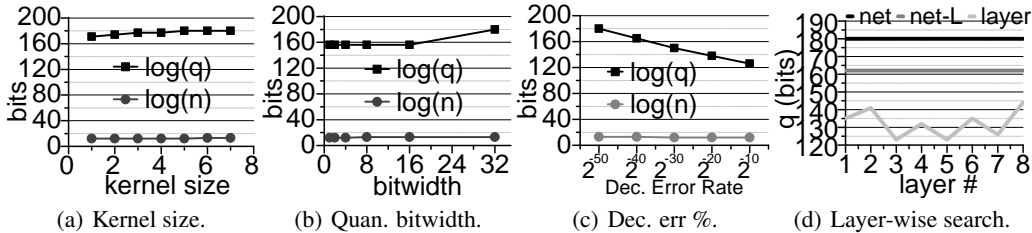

(a) Kernel size.　　(b) Quan. bitwidth.　　(c) Dec. err %.　　(d) Layer-wise search.

Figure 4: The ineffectiveness of conventional neural architecture search.

**Neural Architecture Search**. Deep reinforcement learning (DRL) [15, 16], genetic algorithm [17], and Bayesian optimization [18] are widely used to automatically search a network architecture improving inference accuracy and latency. DRL-found network architectures without privacy-preserving awareness can outperform human-designed and rule-based results [15, 16]. However, naïvely applying DRL on HPPNN architecture search [19] cannot effectively optimize privacy-preserving inference accuracy and latency, because conventional neural architecture search explores the design space of layer number, weight kernel size and model quantization bitwidth, but not HE parameters. Particularly, $n$ and $q$ are not sensitive to changes of weight kernel size, as shown in Figure 4(a). In Figure 4(b), $n$ and $q$ are not sensitive to model quantization bitwidth either, particularly when model quantization bitwidth is < 16. Although smaller weight and bias bitwidths reduce $p$, $p$ has to follow the 5 rules of HE parameters, and thus cannot be reduced in a highly quantized model.

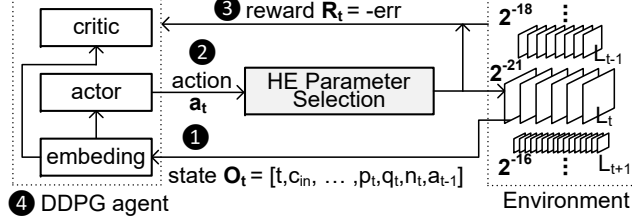

Figure 5: The working flow of AutoPrivacy.

**DRL-based Layer-wise HE Parameter Search**. On the contrary, if the HE decryption error rate is moderately enlarged, as Figure 4(c) shows, $q$ can be obviously reduced. In this paper, as Figure 2(b) shows, we propose a DDPG agent [20], AutoPrivacy, to predict a HE decryption error rate for each layer of a HPPNN to reduce $n$ and $q$, without sacrificing its accuracy. In this way, HPPNN inferences can be accelerated. As Figure 4(d) shows, prior HPPNNs [1, 2, 5, 3] (net) has to select a 60-bit q to guarantee a $> 2^{-40}$ HE decryption error rate for the whole network without considering the error tolerance capability of a neural network. Recent DARL [4] (net-L) finds the upper bounds of HE matrix multiplications, so it can use smaller $q$ but still achieve a $2^{-40}$ HE decryption error rate. However, DARL does not take the error tolerance capability of a neural network into its consideration, nor select a set of $n$ and $q$ for each layer. AutoPrivacy (layer) can choose and minimize $n$ and $q$ for each HPPNN layer by considering its error tolerance capability. As a result, AutoPrivacy greatly decreases HPPNN inference latency without degrading its HE security level or inference accuracy. The search space of selecting a decryption error rate for each layer of a HPPNN is so large that even HE and machine learning experts may obtain only sub-optimal results. There are totally $(D \times S)^{N_L}$, e.g., $\sim 10^8$, options, where $D$ is the number of possible decryption error rates for each layer, e.g., $D = 20$; $S$ is the number of possible HE parameter sets, e.g., $S \approx 5$; and $N_L$ is the layer number of a HPPNN, e.g., $N_L = 8$.

# 3 AutoPrivacy

For each layer in a HPPNN, our goal is to precisely find out the maximal *decryption error rate* that can be tolerated by the layer without degrading the HE security level (128-bit) or the inference accuracy. The HE parameter selection procedure obtains smaller $q$ and $n$ with a higher decryption error rate to shorten the HPPNN inference latency. We first quantize the HPPNN with 8-bit [21] to minimize $p$. Further decreasing the bitwidth of a HPPNN only decreases its accuracy, but cannot further reduce $p$ due to the 5 rules of HE parameters. We formulate the layer-wise decryption error rate prediction task as a DRL problem.

## 3.1 Automated Layer-wise Decryption Error Rate Prediction

As Figure 5 shows, AutoPrivacy leverages a DDPG agent [20] for efficient search over the action space. We introduce the detailed setting of our DDPG framework.

❶ **State Space**. AutoPrivacy considers only linear layers, and thus processes a HPPNN inference layer by layer. For each linear layer $i$ ($L_i$), the state of $O_i$ is represented by $O_i = (i, c_{in}, c_{out}, x_w, x_h, k_s, s_s, p_i, q_i, n_i, a_{i-1})$, where $i$ is the layer index; $c_{in}$ indicates the number of input channels; $c_{out}$ means the number of output channels; $x_w$ is the input width, $x_h$ is the input height; $k_s$ denotes the kernel size; $s_s$ is the stride size; $p_i$ is the plaintext modulus; $q_i$ means the ciphertext modulus; $n_i$ is the polynomial degree; and $a_{i-1}$ is the action in the last time step. If $L_i$ is a fully-connected layer, $O_i = (i, c_{in}, c_{out}, x_w, x_h, k_s = 1, s_s = 0, p_i, q_i, n_i, a_{i-1})$. We normalize each metric in the $O_i$ vector into $[0, 1]$ to make them share the same scale.

❷ **Action Space**. AutoPrivacy uses a HE decryption error rate as action $a_i$ for each linear layer. We adopt a continuous action space to determine the HE decryption error rate. Compared to a discrete action space, the continuous action space maintains the relative order. For example, $2^{-30}$ is more aggressive than $2^{-40}$. For $L_i$, we take the continuous action $a_i \in [0, 1]$, and round it into the discrete HE decryption error rate (DER) $DER_i = 2^{-round(D_l + a_i \times (D_r - D_l))}$, where $2^{-D_l}$ and $2^{-D_r}$ denote the maximal and minimal HE decryption error rate. In this paper, we set $D_l = 5$ and $D_r = 15$. We input the predicted HE decryption error rate to the procedure of HE parameter selection shown in Figure 2(b) to get $p$, $q$, and $n$.

**Latency Constraint on Action Space**. Some privacy-preserving applications have a limited budget on their inference latency. We aim to find the HE parameter policy with the best accuracy under a

latency constraint. We make our agent to meet a given latency budget by limiting its action space. After our agent produces actions for all layers, we measure the HPPNN inference latency with the predicted HE parameters. If the current policy exceeds the latency budget, our agent will sequentially decrease $n$ and/or $q$ of each layer until the latency constraint is satisfied.

**Inference Latency Estimation**. To avoid high HPPNN inference overhead, we profile and record the latencies of polynomial multiplications and additions with various values of $n$ and $q$. From the network topology, we extract key operation information such as the number of homomorphic SIMD multiplications, the number of homomorphic slot rotations, and the number of SIMD additions. The approximate latency of a HPPNN inference can be estimated using the latency and number of each type of operations.

**Inference Accuracy Estimation**. Performing millions of HPPNN inferences on encrypted data is extremely computationally expensive. After AutoPrivacy generates HE parameters for all layers of a HPPNN, instead, we adopt the HE decryption error simulation infrastructure in [4] to estimate the HPPNN inference accuracy. We did not observe any accuracy loss on a HPPNN until the decryption error rate degrades to $2^{-7}$. In most cases, we perform brute-force Monte-Carlo runs. However, to simulate a $2^{-15}$ decryption error rate, at least $2^{30}$ brute-force Monte-Carlo runs are required. To reduce the simulation overhead, we adopt the Sigma-Scaled Sampling [22] to study high dimensional Gaussian random variables. A HE-based linear layer with the initial noise vector $e$ can be abstracted as a function $f(e)$. Its decryption error rate is the probability of the decryption error $\|e\|$ being greater than the noise budget $\eta_t$ generated by HE parameter $p$ and $q$. The decryption error rate can be calculated as $P_d = \int_{-\infty}^{+\infty} I(e)f(e)\mathrm{d}e$, where $I(e) = 1$ if and only if $\|e\| > \eta_t$; otherwise $I(e) = 0$. Sigma-Scaled Sampling reduces the error simulation time by sampling from a different density function $g$, where $g$ is the same as $f$ but scales the sigma of $e$ by a constant $s$. Because $P_g$ offers a much larger probability, we can use less brute-force Monte-Carlo runs to obtain an accurate $P_g$. By scaling factors and model fittings, we can run at most 10 million $P_g$s and convert these values back to $P_d$. We record $\|e\|$s resulting in decryption errors and decompose them by ICRT. We use 50% of the ICRT-decomposed results to retrain the HPPNN by adding them to the output of each linear layer in the forward propagation. And then, we use the other 50% of the ICRT-decomposed results to obtain its inference accuracy.

❸ **Reward**. Since a latency constraint can be imposed by limiting the action space, we define our reward $R$ to be related to only the inference accuracy, i.e., $R = -err$, where $err$ is the HPPNN inference error rate.

❹ **Agent**. AutoPrivacy uses a DDPG agent [20], which is an off-policy actor-critic algorithm for continuous control. In the environment, one step means that the DDPG agent makes an action to decide the decryption error rate for a specific linear layer, while one episode is composed of multiple steps, where the DRL agent chooses actions for all layers. The environment generates a reward $R_i$ and next state $O_{i+1}$. We use a variant form of the Bellman's Equation, where each transition in an episode is defined as $T_i = (O_i, a_i, R_i, O_{i+1})$. During the exploration, the $Q$-function is computed as $\hat{Q}_i = R_i + \gamma \times Q(O_{i+1}, \mu(O_{k+1})|\theta^Q)$, where $\mu()$ is the output of the actor; $Q(,)$ is the output of the critic; $\theta^Q$ is the parameters of the critic network; and $\gamma$ is the discount factor. The loss function can be approximated by $L = \frac{1}{N_s} \sum_{i=1}^{N_s} (\hat{Q}_i - Q(O_i, \mu(O_i)|\theta^Q))^2$, where $N_s$ is the number of steps in this episode.

**Implementation**. The DDPG agent consists of an actor network and a critic network. They share the same network architecture with 3 hidden layers: 400 units, 300 units and 1 unit. For the actor network, we add an additional $sigmoid$ function to normalize the output into range of $[0, 1]$. The DDPG agent is trained with fixed learning rates, i.e., $10^{-4}$ for the actor network and $10^{-3}$ for the critic network. The replay buffer size of AutoPrivacy is 2000. During exploration, the DDPG agent adds a random noise to each action. The standard deviation of Gaussian action noise is initially set to 0.5. After each episode, the noise is decayed exponentially with a decay rate of 0.99.

**Finetuning**. During exploration, we finetune the HPPNN model with generated decryption errors for one epoch to recover the accuracy. We randomly select 2 categories from CIFAR-10 (10 categories from CIFAR-100) to accelerate the HPPNN model finetuning during exploration. After exploration, we generate decryption errors based on the best HE parameter selection policy and finetune it on the full dataset.

## 4    Experimental Methodology

We performed extensive experiments to show the consistent effectiveness of AutoPrivacy to minimize the HPPNN inference latency with trivial loss of accuracy.

**Hardware configuration**. We ran HPPNN inferences and measured the latency of each type of operations on an Intel Xeon E7-4850 CPU with 1TB DRAM. We assume the same network LAN setting as DELPHI [5]. We implemented and trained AutoPrivacy on a Nvidia GTX1080-Ti GPU.

**HE and GC setting**. We implemented HE-based linear layers of HPPNNs by Microsoft SEAL library [6], and GC-based nonlinear layers of HPPNNs through the swanky library [23]. Because we quantized all network models with 8-bit, we fix the plaintext modulus $p$ as 14 [4]. To evaluate the security level of a set of HE parameters, we relied on the LWE-Estimator [12]. The same as DELPHI [5] and DARL [4], all HE parameters we studied satisfy the $> 128$-bit security level. To estimate the inference accuracy, we use the original HE parameters $n$ and $q$. On the contrary, we use $4 \times n$ and $3 \times \log(q)$ to enable noise flooding and evaluate inference latency.

**Dataset and model**. Our experiments are performed on the CIFAR-10/100 dataset. We studied a series of neural network architecture including a 7-layer CNN network used by [5] (7CNET), ResNet-32 [24] (RESNET), and MobileNet-V2 [25] (MOBNET). 7CNET consists of seven convolutional layers. MOBNET consists of pointwise and depthwise convolution layers, each of which is a pointwise-depthwise-pointwise block. Only 7CNET is trained and tested on CIFAR-10, while experiments of RESNET and MOBNET are performed on CIFAR-100.

| Network | Scheme | Latency (s) | | | Communication (GB) | | | accuracy (%) |
|---|---|---|---|---|---|---|---|---|
| | | $t_{off}$ | $t_{on}$ | $t_{total}$ | $m_{off}$ | $m_{on}$ | $m_{total}$ | |
| 7CNET | DELPHI | 41 | 0.8 | 41.8 | 0.12 | 0.01 | 0.13 | 81.63 |
| | DARL | 28.7 | 0.42 | 29.12 | 0.11 | 0.01 | 0.12 | 81.63 |
| | **AutoPrivacy** | 10.24 | 0.31 | **19.55** | 0.09 | 0.01 | **0.1** | **81.5** |
| RESNET | DELPHI | 90 | 6.4 | 96.4 | 1.9 | 0.04 | 1.94 | 76.78 |
| | DARL | 56.7 | 3.83 | 60.53 | 1.7 | 0.04 | 1.74 | 76.78 |
| | **AutoPrivacy** | 27 | 1.56 | **28.56** | 1.07 | 0.03 | **1.1** | **76.78** |
| MOBNET | DELPHI | 17.4 | 1.74 | 19.14 | 0.24 | 0.01 | 0.25 | 68.08 |
| | DARL | 11.2 | 1.23 | 12.43 | 0.23 | 0.01 | 0.24 | 68.08 |
| | **AutoPrivacy** | 6.2 | 0.72 | **6.92** | 0.19 | 0.01 | **0.2** | **68.05** |

Table 2: The execution time, communication overhead and inference accuracy comparison.

## 5    Results and Analysis

**Overall Performance**. The execution time, communication overhead, and inference accuracy comparison between prior HPPNNs and AutoPrivacy-optimized HPPNNs are shown in Table 2. Compared to DELPHI, our AutoPrivacy-optimized counterparts reduce the inference latency by $53\% \sim 70\%$, decrease the ciphertext size by $20\% \sim 43\%$, and maintain a trivial inference accuracy loss (0.1%). Particularly, compared to RESNET, AutoPrivacy reduces the offline inference latency by 70%, and the online inference latency by 75%. If a client infer multiple images, only the first one costs 28.56 seconds. It takes only 1.56 seconds for each of the other images tested by a heavyweight RESNET model. The CRT, ICRT, NTT and RNS processing operations during HPPNN inferences are greatly accelerated by the HE parameters automatically selected by AutoPrivacy. Although smaller $q$ and $n$ may generate more decryption errors, HPPNNs naturally tolerate most errors without obviously decreasing inference accuracy. We observe only 0.1% accuracy loss for MOBNET and 7CNET. Finetuning is critical to recover the inference accuracy degradation caused by smaller HE parameters $q$ and $n$. We find on average finetuning improves inference accuracy by 8%. Especially, finetuning can eliminate the accuracy loss for RESNET.

**HE Parameter Selection**. We report the details of HE parameter selection of RESNET and MOBNET inferring on the CIFAR-100 dataset in Figure 6(a) and (b) receptively. For CIFAR-100, besides the first convolutional layer and the last fully-connected layer, RESNET applies a stack of $6M$ layers with $3 \times 3$ convolutions on the feature maps of sizes of $\{32, 16, 8\}$ respectively on $32 \times 32$ images, where $M$ is an odd integer. $2M$ layers for each feature map size form a residual block. As Figure 6(a) shows, AutoPrivacy automatically observes the boundary of each residual block of RESNET. Inside each residual block, AutoPrivacy identifies the 2nd and 6th layers can work with smaller $q$ and $n$, since they have less influence to the inference accuracy. On the contrary, the 4th and 8th layers in a residual

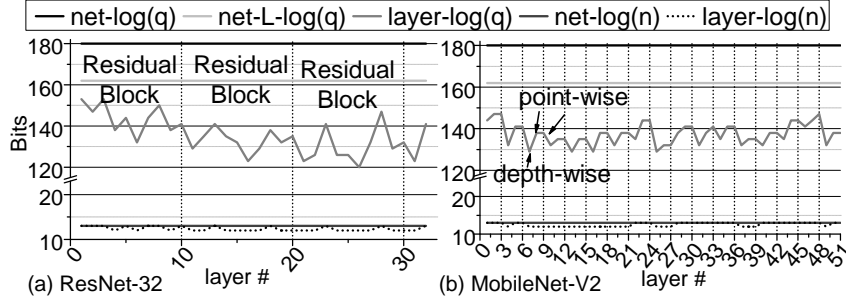

Figure 6: The HE parameter comparison of AutoPrivacy against prior works (*net*-log*(q)* and *net*-log*(n)* indicate the $q$ and $n$ of DELPHI, *net-L*-log*(q)* and *net*-log*(n)* mean the $q$ and $n$ of DARL, and *layer*-log*(q)* and *layer*-log*(n) denote the q and $n$ of AutoPrivacy*).

block have to use larger $q$ and $n$, because they own larger weights in determining the inference accuracy. For MOBNET, AutoPrivacy automatically finds the difference between depth-wise and point-wise convolutions. Depth-wise convolution layers have less accumulations thereby reducing the number of HE rotation operations that greatly increase the noises in packed ciphertexts. Therefore, AutoPrivacy assigns smaller $n$ and $q$ to depth-wise convolution layers without sacrificing the inference accuracy. In contrast, point-wise convolution layers have $1 \times 1$ convolutions and tens to hundreds of output channels requiring a great number of accumulations. Point-wise convolution layers have to invoke many HE rotation operations in ciphertexts, and thus increase HE noises in ciphertexts. To tolerate larger HE noises, AutoPrivacy has to select larger $n$ and $q$ to provide larger noise budgets in point-wise convolution layers without human guidance.

| Name | Network Architecture | Quantization (bits) | Latency (seconds) | Communication(GB) | Accuracy (%) | Search time (hours) |
|------|----------------------|---------------------|-------------------|-------------------|--------------|---------------------|
| NASS [19] | 5 CONV. + 1 FC | 4~16 | 20.1 | 0.978 | 84.6 | 60 |
| AutoPrivacy | MOBNET | 14 | 6.13 | 0.2 | 91.4 | 8 |

Table 3: The comparison between NASS and AutoPrivacy.

**Comparison against NASS**. A recent work, NASS [19], automatically builds a privacy-preserving neural network architecture by a deep reinforcement learning agent. However, instead of HE parameters, NASS automatically searches neural network architectures and quantization bitwidths for each linear and nonlinear layer. As a result, its search space size is too large to be efficiently and effectively explored. Table 3 highlights the comparison of results achieved by NASS and AutoPrivacy searching for the CIFAR-10 dataset. NASS finds a network architecture with five convolutional layers and one fully-connected layer on the CIFAR-10 dataset. It also quantizes each linear and nonlinear layers with $4 \sim 16$ bits. On the contrary, we train a MOBNET on the CIFAR-10 dataset and quantize the model with 14-bit. Compared to the NASS-found network, MOBNET optimized by AutoPrivacy improves the inference latency by 69.5%, the communication overhead by 79%, and the inference accuracy by 8%. The search of AutoPrivacy takes only 8 hours, but the search time of NASS is $> 60$ hours. This is because each time NASS has to train a neural network from scratch, then quantize it, and finally retrain it, once it selects a topology for the HPPNN. The design space is too large for its deep reinforcement learning agent. In contrast, we argue that the emerging compact network architectures like MOBNET can maximize the inference accuracy with less parameters. We can use a pre-decided network architecture, quantize it with the same bitwidth, and rely on AutoPrivacy to automatically choose HE parameters for each linear layer of the fixed architecture. Compared to the network architecture and quantization bitwidth, choosing appropriate HE parameters for linear layers of the fixed network more effectively reduces the inference latency.

## 6   Conclusion

In this paper, we propose, AutoPrivacy, an automated layer-wise HE parameter selector to optimize for fast and accurate HPPNN inferences on encrypted data. AutoPrivacy uses a deep reinforcement learning agent to automatically find a set of HE parameters for each linear layer in a HPPNN without sacrificing the 128-bit security level. Compared to prior HPPNNs, AutoPrivacy-optimized HPPNNs reduce the inference latency by $53\% \sim 70\%$ with a negligible accuracy loss.

## Broader Impact

In this paper, we propose an automated HE parameter selector for non-experts, i.e., average users, to automatically optimize their privacy-preserving neural network, so that average users can work with fast and accurate privacy-preserving neural network inferences on encrypted data. Average users, who have to rely on big data companies but do not trust them, can benefit from this research, since they can upload only their encrypted data to untrusted servers. No one may be put at disadvantage from this research. If our proposed technique fails, everything will go back to the state-of-the-art, i.e., untrusted servers may leak sensitive data of average users.

## Acknowledges

The authors would like to thank the anonymous reviewers for their valuable comments and helpful suggestions. This work was partially supported by the National Science Foundation (NSF) through awards CCF-1908992 and CCF-1909509. Song Bian was partially supported by JSPS KAKENHI Grant No. 20K19799.

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
