[Reviews · NeurIPS 2020]

Review 1

Summary and Contributions: The paper describes a new method for selecting Homomorphic Encryption scheme parameters used for Linear layers in a secure inference setting. The authors propose using a DRL agent to select these optimal parameters on a layer by layer basis. The authors show up to 70% reduction in inference latency compared to state-of-art using their new technique

Strengths: The main strength of the paper is its critical analysis of the requirement on HE schemes to provide very strong decryption correctness guarantees (< 2^{-40} decryption error rate) assumed by previous. The authors describe experiments showing that observed negligible decrease in end-to-end accuracy in the presence of decryption error rates as high as 2^{-7}. This allows the authors to significantly relax the HE parameters used resulting in improved latency.

Weaknesses: The authors propose a DRL algorithm to select the exact HE parameters on a layer by layer basis. In practice the DRL agent selects just two parameters q and n. Of these two parameters it seems that the practical search space for n is one of three (?) possible values (Fig 6). The range for q also shows about a 20% variation. Since CRT based packing allows for nearly free parallelization the key metric of interest should be the latency per slot. The dependence of latency per slot on n is only logarithmic for slot rotation and independent of n for addition and multiplication. The dependency on q depends on the number of CRT limbs required (which three for most layers in Fig 6). In this context it is not clear how a 70% reduction in runtime is achieved

Correctness: There are three main concerns regarding the correctness of the authors conclusions: 1. It is not clear whether the DRL agent offers any significant advanatage over just a fixed aggressive setting of HE parameters for all layers. 2. It is not clear what exact models were in predict network latency since there are significant quantization issues in practice e.g. computation runtime scaling with number of limbs vs bitwidth of q. 3. Fig 3. does not fairly represent the increased runtime per slot with changing n.

Clarity: The paper is written fairly clearly. Some of the figure are a bit cluttered but perhaps that is inevitable due to length restrictions. There is perhaps a typo on line 120 where n is mentioned to be > 10^{10}. Also the citations for Delphi are messed up with those for Gazelle in multiple locations

Relation to Prior Work: The authors clearly highlight that their contribution differs from prior art in the more agressive setting for decryption error rates (which they state has a negligble impact on the inference accuracy). The also highlight their use of a DRL agent for setting HE parameters as different from prior art.

Reproducibility: Yes

Additional Feedback:


Review 2

Summary and Contributions: The paper investigates the trade-off between errors and efficiency when using homomorphic encryption for the linear layers in privacy-preserving neural network inference. It presents an automated approach to choose these parameters such that the overall computation remains correct while allowing partial errors. It also present concrete results that show an up three times lower latency while preserving overall correctness.

Strengths: The paper explores an interesting combination of the particulars of homomorphic encryption and deep neural networks, and it provides concrete improvements. The idea that cheaper HE parameters lead to errors in a few linear layer outputs but globally makes intuitive sense considering that non-linear layers are computed error-free. I have not seen a treatment of this issue in this depth previously.

Weaknesses: The consideration of HE parameters only seems to consider ciphertexts before multiplication. Post rebuttal: I do acknowledge the authors' focus on an empirical error analysis, but I still feel that the treatment of HE is a hap-hazardous.

Correctness: The homomorphic encryption is recognisable and the overall approach to parameter finding is in line with previous works, but: - The parameter consideration seems insufficient. No space is given to the correct decryption of multiplied ciphertexts, which are more relevant than the unmultiplied ciphertext in Equations 1 and 2. - The formulation of BFV does not exactly match the cited previous works. - Figure 1.d) is insufficiently explained. Why does n not influence the the accuracy?

Clarity: The paper is written clearly but grammar and style could be improved, for example: l22: "Privacy is especially important(,) when" l65: The causality of this sentence makes no sense. l101: "e0 can be rounded off" l106: |\gamma| is an integer and it makes little sense for it to be roughly 1 in this context. l107: "adopts (the) Chinese Reminder Theorem" l123: "prior NPPNNs compute(s)" l139: "use (the) Residue Number System" l141: "As Figure 3.2 (shows)" l167: "that (a) HPPNN inference can be accelerated" l178: "so huge" l198: "is (a) fully-connected layer" l202: "the continuous action space maintain(s) the relative order" l208: "privacy-persevering" l218: "we can fast calculate" l221: "err is (the) HPPNN inference error rate" l226: "we perform brute-force Monte-Carlo runs are required" l246: "(the) Q-function is computed" l270: "through (the) swanky library" l278: "7CNet consisting (consists) of" l290: "each of the following images" - Which images? l308: "AutoPrivacy automatically find(s) the" l348: "may leak(age) sensitive data" several times: what are NPPNNs? HPPNNs? References: capitalization of "lwe" and "gapsvp"

Relation to Prior Work: Yes.

Reproducibility: Yes

Additional Feedback:


Review 3

Summary and Contributions: This paper propose an automated layer-wise parameter selector, AutoPrivacy, that leverages deep reinforcement learning to automatically determine a set of HE parameters for each linear layer in a HPPNN. The target is fast and accurate secure neural network inference. It reduces inference latency by 53% ∼ 70% with negligible loss of accuracy.

Strengths: The empirical evaluation of the latency is thorough.

Weaknesses: The motivation why using RL to search the hyper-parameter for secure neural network inference is not discussed. The author didn't compare with other hyper-parameter search algorithms such as Bayesian optimization, evolutionary strategy, even random search. The sample efficiency is also not discussed in the paper either.

Correctness: - 19.55 seconds of inference latency on the Cifar dataset isn't practically useful.

Clarity: It's readable.

Relation to Prior Work: Similar techniques have been widely explored in many neural architecture search papers since 2018. This paper is a combination of existing approaches but targeting a different problem.

Reproducibility: No

Additional Feedback: - compare with Bayesian optimization, evolutionary strategy, and random search - compare the sample efficiency and reproducibility with other hyper parameter optimization methods - provide evidence of the reproducibility of the DDPG RL agent by showing the learning curve in multiple runs; show the search time and GPU hours.


Review 4

Summary and Contributions: This paper studied the private deep neural network questions. HPPNN is a method that implements linear layers by Homomorphic encryption and nonlinear layers by Garbled Circuit. This work propose an automated layer-wise parameter selector, that leverages deep reinforcement learning to automatically determine a set of HE parameters for each linear in a HPPNN.

Strengths: From experiments side, this new method reduces inferences latency by 53%~70% with negligible loss of accuracy.

Weaknesses: This paper seems to be an experimental paper, but no theory. It is probably ok.

Correctness: Yes

Clarity: I am a theory person, not doing experiments much in general. From my perspective, the paper is well-written.

Relation to Prior Work: Some related citations are missing, if the paper is got accepted, these works should be discussed in the related work in the camera-ready version. [1] Privacy-preserving Learning via Deep Net Pruning Yangsibo Huang, Yushan Su, Sachin Ravi, Zhao Song, Sanjeev Arora, Kai Li [2] InstaHide: instance-hiding schemes for private distributed learning Yangsibo Huang, Zhao Song, Kai Li, Sanjeev Arora

Reproducibility: Yes

Additional Feedback:

[Author Response · NeurIPS 2020]

We thank all reviewers for their careful reading of the manuscript and their constructive comments.

**Reviewer-1: The search space of** $n$. In Figure 2(b), $log(n)$ has to be an integer, which is $9 \leq log(n) \leq 14$.

**Reviewer-1: The latency per slot**. Instead of the latency per slot, we reported the end-to-end inference latency of a
privacy-preserving neural network as the key performance metric. First, we would like to emphasize that the numbers
of $n$ and $q$ we provided in Figure 3 and 6 are **all in the log scale**. We also would like to point out that the comment
of "The range for $q$ also shows about a 20% variation" is **inaccurate**. What we have is "The range for $log(q)$ also
shows about a 20% variation". Second, although we agree with the reviewer on the analysis on the latency per slot, we
think the reviewer misunderstood the latency per slot on our baseline using the same $n$ and $q$ for all layers. Different
neural network layers have different numbers of input and output channels, and different weight kernel sizes. Therefore,
different layers in a neural network require different values of $n$ to pack all weights. Our baseline selected a large $n$ to
have enough slots to pack the layer with the largest number of weights. However, for the other layers, most slots are
empty since they have less weights. In this way, the "effective" latency per slot of our baseline is very long, where the
effective latency means the latency per non-empty slot.

**Reviewer-1: The 70% latency reduction**. We would like to emphasize the fact that the numbers of $n$ and $q$ we
provided in Figure 3 and 6 are **all in the log scale** again. We measured the latency of each HE layers on a real machine.
By reducing $n$ and $q$, the cache hit rate greatly increases during the NTT and CRT computations. Therefore, we
observed a great latency reduction. We do NOT think simply scaling the latency per slot with $n$ and $q$ is a good latency
estimation.

**Reviewer-1: Comparison against the fixed aggressive setting**. We compared AutoPrivacy against DARL [4] in Table
2 and Figure 6. DARL aggressively sets the same $n$ and $q$ for all layers of a neural network.

**Reviewer-1: The models were used**. We explained the models we studied in Section 4. We studied a 7-layer CNN
network used by [5] (7CNET), ResNet32 [24] (RESNET), and MobileNet-V2 [25] (MOBNET). We quantized all
models with 8-bit. Due to the limited space, we cannot include the details of the network architecture in the manuscript.
We will try to add the information in the next version of this manuscript.

**Reviewer-1: Figure 3 does not represent the runtime per slot**. Figure 3 shows the execution time of a HE multipli-
cation we measured on a real machine with different values of $n$ and $q$. Again, we believe the reviewer underestimated
the "effective" latency per slot of our baseline.

**Reviewer-2: Analysis on the decryption of multiplied ciphertexts**. We believe it is difficult to do a mathematical
analysis on the error rate of a neural network with different values of $n$ and $q$. A mathematical error derivation is too
complicated for an inference of a specific privacy-preserving neural network. This is why we propose AutoPrivacy
that selects a set of $n$ and $q$, and feeds them into the HE protocol and real HE-enable neural network to calculate the
accuracy. The error tolerance of a privacy-preserving neural network is architecture- and application-dependent.

**Reviewer-2: Figure 1(d) Why** $n$ **does not influence the accuracy**. The decryption error is not related to $n$. $n$ decides
the number of slots that can be packed in a cyphertext. Any large convolution can be broken into smaller pieces, so that
we can always use a smaller $n$ to perform the computation but with longer latency.

**Reviewer-3: Comparison against the other search techniques**. In this paper, we present a new and important prob-
lem on the parameter selection of privacy-preserving neural networks. Presenting the problem is our first contribution.
Our design is the first work to identify the fact that the mathematical error derivation is not necessary for the inferences of
a privacy-preserving neural network. We can achieve better inference latency than two of the most recent state-of-the-art
designs. We will compare AutoPrivacy against other search techniques in the next version of this manuscript.

**Reviewer-3: 19.55 seconds of inference latency on the Cifar dataset isn't practically useful**. The inference time is
architecture-dependent, but not dataset-dependent. By the architecture of 7CNNet, our fastest inference on CIFAR-10
requires only 6.92 seconds, which is much faster than all existing state-of-the-art designs.

[Meta-Review · NeurIPS 2020]

The paper describes a new method for selecting Homomorphic Encryption scheme parameters used for Linear layers in a secure inference setting. The paper explores an interesting combination of the particulars of homomorphic encryption and deep neural networks and is able to provide strong decryption correctness guarantees.